# Synthesis of Densely Immobilized Gold-Assembled Silica Nanostructures

**DOI:** 10.3390/ijms22052543

**Published:** 2021-03-03

**Authors:** Bomi Seong, Sungje Bock, Eunil Hahm, Kim-Hung Huynh, Jaehi Kim, Sang Hun Lee, Xuan-Hung Pham, Bong-Hyun Jun

**Affiliations:** 1Department of Bioscience and Biotechnology, Konkuk University, Seoul 143-701, Korea; iambomi33@konkuk.ac.kr (B.S.); bsj4126@konkuk.ac.kr (S.B.); greenice@konkuk.ac.kr (E.H.); huynhkimhung82@gmail.com (K.-H.H.); susia45@gmail.com (J.K.); bjun@konkuk.ac.kr (B.-H.J.); 2Department of Chemical and Biological Engineering, Hanbat National University, Daejeon 34158, Korea; sanghunlee@hanbat.ac.kr

**Keywords:** gold nanostructure, dense gold-assembled silica nanostructures, local surface plasmon resonance, peroxidase-like catalysis, surface enhanced Raman scattering

## Abstract

In this study, dense gold-assembled SiO_2_ nanostructure (SiO_2_@Au) was successfully developed using the Au seed-mediated growth. First, SiO_2_ (150 nm) was prepared, modified by amino groups, and incubated by gold nanoparticles (ca. 3 nm Au metal nanoparticles (NPs)) to immobilize Au NPs to SiO_2_ surface. Then, Au NPs were grown on the prepared SiO_2_@Au seed by reducing chloroauric acid (HAuCl_4)_ by ascorbic acid (AA) in the presence of polyvinylpyrrolidone (PVP). The presence of bigger (ca. 20 nm) Au NPs on the SiO_2_ surface was confirmed by transmittance electronic microscopy (TEM) images, color changes to dark blue, and UV-vis spectra broadening in the range of 450 to 750 nm. The SiO_2_@Au nanostructure showed several advantages compared to the hydrofluoric acid (HF)-treated SiO_2_@Au, such as easy separation, surface modification stability by 11-mercaptopundecanoic acid (R-COOH), 11-mercapto-1-undecanol (R-OH), and 1-undecanethiol (R-CH_3_), and a better peroxidase-like catalysis activity for 5,5′-Tetramethylbenzidine (TMB) and hydrogen peroxide (H_2_O_2_) reaction. The catalytic activity of SiO_2_@Au was two times better than that of HF-treated SiO_2_@Au. When SiO_2_@Au nanostructure was used as a surface enhanced Raman scattering (SERS) substrate, the signal of 4-aminophenol (4-ATP) on the surface of SiO_2_@Au was also stronger than that of HF-treated SiO_2_@Au. This study provides a potential method for nanoparticle preparation which can be replaced for Au NPs in further research and development.

## 1. Introduction

Metal nanoparticles (NPs) have attracted much attention due to their quantum size effect resulting in unique physical and chemical properties [1]. Among metal nanostructures, gold (Au) nanostructures are probably the most remarkable metal materials due to their particular features: ease of synthesis manipulation; large surface-to-volume ratio, enabling precise control over the particle’s physicochemical properties [2]; strong binding affinity to thiols, disulfides, and amines [3]; unique tunable optical properties due to their size- and shape-dependent biophysical and distinct optoelectronic properties [4,5,6,7]; excellent biocompatibility, chemical inertness, and low toxicity [8,9]. As a result, Au nanostructures have been outstanding tools for a variety of potential applications in catalysis [10,11,12,13,14,15] and biosensing [16,17,18,19,20,21,22].

Based on their dimensions, Au nanostructures are classified as one-dimensional (1D) nanostructures (nanorods, nanowires, nanotubes, nanobelts); two-dimensional (2D) nanostructures (nanoplates such as nanoparticles, stars, pentagons, squares/rectangles, dimpled nanoplates, hexagons, truncated triangles); and three-dimensional (3D) nanostructures (nanotadpoles, nanodumbbells, nanopods, nanostars, and nanodendrites) [23]. Even though spherical or quasi-spherical Au NPs have received the most attention because of the ease of synthesis manipulation, anisotropically Au nanostructures exhibit unique physical and optical properties, such as low percolation threshold and surface plasmon resonance (SPR) [24]. In addition, the optical properties of anisotropic Au nanostructure such as the hollow nanoshell, and nanorods can be tunable with their shape in the visible region and in the near infrared (NIR) region. This property has given rise to the opportunity for using anisotropic Au nanostructures as composite therapeutic agents in clinical medical applications such as diagnostics and therapy [25]. However, the synthesis of higher-dimensional Au nanostructures usually requires specific templates for guiding anisotropic growth [26]. Many templates have been utilized, including biomolecules (DNA), polymers, surfactants, inorganic nanowires, and lithographical patterns [27,28,29,30,31]. However, the synthesis of anisotropic Au nanostructures with precise control of morphology remains a great challenge.

Recently, our group reported Au–Ag alloys assembled SiO_2_ (SiO_2_@Au–Ag) nanostructure as a strong and reliable surface enhanced Raman scattering (SERS) probe using SiO_2_ as a template [32,33,34,35]. SiO_2_ NPs are inert and their sizes are easily controllable [32]. Furthermore, the density of Au NPs, nanogaps, size and shape of nanostructure can be controlled by SiO_2_ template, generating a homogenous SERS substrate [36]. The density of the Au NPs and the gaps between two Au NPs on the surface of SiO_2_ can be tunable and provide a stronger localized surface plasmon resonance (LSPR) property. As a result, SiO_2_@Au–Ag can be developed as high-strength and reliable SERS substrates for the enzyme-linked immunosorbent assay (ELISA) [37,38], which is an internal SERS [34,35], pesticide detection standard [39], and provides detection of toxic chemical compound in pharmaceutics [40]. However, the cellular toxicity and easy oxidation of intrinsic Ag of our nanostructure limits their application in vivo [41,42]. Even though some research groups have investigated the attachment of Au NP on the surface of SiO_2_, the particle density on the SiO_2_ surface was nearly low or non-uniform [43,44,45,46,47,48]. Therefore, we used the combination of SiO_2_ core template and seed-growth methods in this study to develop a new Au-based SiO_2_ nanostructure with dense Au density on the surface of SiO_2_. This study provides a potential method for preparation of nanoparticles including noble metal that can be replaced for spherical Au NPs, which supports further research and development.

## 2. Results and Discussion

To prepare SiO_2_ nanostructure (SiO_2_@Au), silica NPs (~150 nm) were first functionalized by 3-aminopropyltriethoxysilane (APTS) to prepare aminated silica NPs, as shown in Figure 1a. Simultaneously, colloidal Au NPs (3 nm) were synthesized by tetrakis(hydroxymethyl)phosphonium chloride (THPC) and incubated with the aminated silica NPs to prepare Au NPs seed embedded with SiO_2_ (SiO_2_@Au seed NPs), according to the method reported by Pham et al. [32,33,34,35,39]. Subsequently, the Au NPs on the surface of SiO_2_@Au seed were grown by reducing a gold precursor (HAuCl_4_) in the presence of ascorbic acid (AA) and polyvinylpyrrolidone (PVP) as a stabilizer and structure-directing agent under mild reducing conditions [32]. The gold ions reduced by AA were selectively grown onto SiO_2_@Au seed to increase the size of Au on the surface of SiO_2_ NPs to generate SiO_2_@Au.

### 2.1. Synthesis of SiO_2_@Au Nanostructure

We investigated the characteristics of SiO_2_@Au synthesized by reduction of HAuCl_4_ and AA in the presence of PVP. Figure 1b shows the optical images of nanostructures in our research. SiO_2_ NPs with ca. 150 nm in diameter are shown in a transparent solution in Figure 1b(i). When Au NPs were coated and grown on the surface of aminated SiO_2_, the color of the solution changed to light brown, as shown in Figure 1b(ii). Then, 2, 5, and 10 mg of SiO_2_@NH_2_ were incubated in 10 mL Au NPs suspension overnight (Appendix A). The surface of SiO_2_ NPs was decorated with many small Au NPs to generate the SiO_2_@Au seed (Figure 1b(ii) and Appendix A). The Au NPs density on the SiO_2_ surface became inversed with the amount of SiO_2_ used. Then, 1 mg of SiO_2_@Au seed was used for a growth of Au NPs on the SiO_2_ in Appendix A. The sizes of Au NPs seed on the SiO_2_@Au surface increased with the Au^3+^ concentration in the range of 100 to 300 µM at all 2, 5, and 10 mg SiO_2_@NH_2_ (Appendix A). The ultraviolet-visible (UV-Vis) spectra of obtained SiO_2_@Au nanostructures synthesized at different concentrations of Au^3+^ were investigated in Appendix A. The optical property of SiO_2_@Au was slightly tunable by adjusting the Au^3+^ concentration in the range of 100 to 300 µM. At 100 µM Au^3+^, the maximum absorbance peaks of obtained SiO_2_@Au synthesized by 2, 5, and 10 mg SiO_2_@NH_2_ were observed at 523, 522, and 521 nm, respectively. Whereas these peaks were red-shifted to 547, 543, and 541 nm at 200 µM Au^3+^ and 571, 561, and 572 nm at 300 µM Au^3+^, respectively. These results were consistent with the Mie’s theory which states that an increase of particle size leads the electric surface charge density and shifts the plasmon absorption band to longer wavelengths [23]. However, it is difficult to obtain the different desired SiO_2_@Au nanostructures as expected when using 1 mg SiO_2_@Au seed. The final size of Au NPs is dependent on the number of seeds and total concentration of Au^3+^ in the growth solution [49]. In our study, the Au^3+^ concentration in the growth solution was fixed at 100, 200, and 300 µM. Therefore, the final size of Au NPs on the surface of SiO_2_ depended on the number of SiO_2_@Au seeds (or SiO_2_@Au seed amount). Indeed, 1 mg of SiO_2_@Au seed was used for Au NPs growth process in our study, which resulted in Au NPs on the SiO_2_ surface not being able to grow larger as seen in the TEM images and UV-Vis spectra (Appendix A). To obtain a monodisperse SiO_2_@Au nanostructure, the SiO_2_@Au seed was therefore decreased to 0.2 mg for the growth of Au NPs on the SiO_2_ NPs in the next study. When 0.2 mg SiO_2_@Au seed was used as seed, the color of the obtained SiO_2_@Au suspension obviously changed from light brown to dark blue when the Au^3+^ concentration increased to 150 µM, as shown in Figure 1b(iii). According to the literature, Au NPs exhibit color and LSPR bands in the visible light that are dependent on size and shape [49,50,51,52]. Any changes in the absorbance or wavelength of the LSPR band provide a difference in size, shape, and aggregation state. This result indicated that the growth of Au NPs on the SiO_2_ surface was well performed and the absorbance, transmission, and reflection of light passed through the SiO_2_@Au suspension were also different at various synthesis conditions of SiO_2_@Au nanostructures. The UV-Vis spectra of the SiO_2_@Au nanostructures synthesized at 150 µM Au^3+^ were investigated in Figure 1c to show additional characteristics of SiO_2_@Au nanostructures. The suspension of SiO_2_ and SiO_2_@Au seed did not show any absorbance in the range of 300–1000 nm because of its small Au NPs on the SiO_2_ surface [37,53]. Whereas the suspension of SiO_2_@Au synthesized at 150 µM Au^3+^ showed a broadband in the range of 450 to 750 nm with the maximum peak at 615 nm. Furthermore, the absorbance intensity of SiO_2_@Au nanostructure solution was sharply increased. Consistent with the UV-Vis spectra, the transmittance electronic microscopy (TEM) images of SiO_2_@Au nanostructures also showed that the size of Au NPs on the surface of SiO_2_@Au increased dramatically comparing to the SiO_2_@Au seed at 150 µM Au^3+^ (Figure 1d), which led to the sharp broadening of LSPR bands of UV-Vis spectrum. While the average size of SiO_2_@Au seed was 200 ± 9 nm (*n* = 170, Appendix A), the average size of SiO_2_@Au was 211 ± 7 nm (*n* = 172, Appendix A). Consequently, the red-shifted and broadened absorbance properties of SiO_2_@Au nanostructures indicated the growth of Au NPs to bigger size on the SiO_2_ surface [23,49]. Therefore, the SiO_2_@Au nanostructure synthesized at 150 µM Au^3+^ was used for further study.

### 2.2. Characteristics of SiO_2_@Au Nanostructure

As shown in the previous study, the presence of SiO_2_ template in SiO_2_-based nanostructure facilitates the control of density, size, shape, and nanogap of metal NPs on the SiO_2_ surface [36]. It is believed that SiO_2_ template also facilitates the separation of SiO_2_-based nanostructures in reaction compared to those without SiO_2_ template. In our study, SiO_2_@Au nanostructure was first obtained from the growth reaction at 150 µM Au^3+^, then was incubated in 24% hydrofluoric acid (HF) to etch the SiO_2_ core template as seen in Figure 2a. Figure 2b shows an obvious color change from blue to pink when the SiO_2_ template was etched by HF. The TEM images of SiO_2_@Au nanostructure with HF treatment were collected. Figure 1d(iii) shows the morphology of SiO_2_@Au without HF treatment with the average size of 211 ± 7 nm (Appendix A). Many Au NPs presented on the surface of SiO_2_ template. However, the nanostructure with HF treatment showed many small Au NPs with different shape (Figure 2c(ii)). The average size of Au NPs was 7.7 ± 1.5 nm (*n* = 172, Appendix A). This meant that SiO_2_ core was completely etched and disappeared in SiO_2_@Au nanostructure and remained as small Au NPs. The optical properties of SiO_2_@Au with and without HF treatment (Au) were significantly different, as shown in Figure 2d. Both the maximum peak position and absorbance intensity of SiO_2_@Au with HF treatment (Au) decreased dramatically. The maximum peak position of SiO_2_@Au at 615 nm was blue-shifted to 546 nm when SiO_2_ core was etched. The difference in the LSPR wavelength and absorbance intensity of Au (SiO_2_@Au with HF treatment) was due to the difference in aggregation state of nanostructure that was caused by the disappearance of SiO_2_ core template [49]. To confirm the aggregation state of SiO_2_@Au with and without SiO_2_ core, the nanostructures were redispersed in 1 M NaCl. The maximum LSPR peak of SiO_2_@Au in NaCl showed a 10 nm red-shift compared to it in distilled water. Whereas SiO_2_@Au treated with HF was 22 nm red-shifted in 1 M NaCl (Appendix A). This indicates that SiO_2_@Au treated with HF was more easily aggregated than SiO_2_@Au without HF treatment. 

The separation and redispersion of SiO_2_@Au nanostructures was also observed in Figure 2f,g and Appendix A. The SiO_2_@Au with and without HF treatment were centrifuged at various centrifugation speeds in the range of 1000 to 18,000× *g*. The SiO_2_@Au was almost settled down to a bottle of microtube at a centrifugal speed of 3000× *g* and left a supernatant which showed a low absorbance. Meanwhile, the absorbance intensity of the supernatant of SiO_2_@Au treated with HF decreased gradually to 50% at 1000× *g*, 25% at 3000× *g* and 6.3% at 10,000× *g*. From these results, we concluded that SiO_2_@Au was easily centrifuged and separated from the reaction solution. In the next experiment, we carried out centrifuge and successively redispersed the SiO_2_@Au at the centrifugation speed of 3000× *g*. As seen in Figure 2g and Appendix A, both SiO_2_@Au with and without HF treatment was easily redispersed in phosphate buffer saline containing 0.1% Tween 20 (PBST). Therefore, we concluded that the SiO_2_@Au nanostructures were easily separated and redispersed in PBST at mild conditions. The use of SiO_2_ as the core template opens up a new opportunity for easy preparation and separation of the nanostructure. 

### 2.3. Surface Modification of SiO_2_@Au Nanostructure

According to the literature, the electric field of the incident light induces polarization and excites the free conduction electron on the surface of the nanostructure to generate the LSPR spectrum. Therefore, any variation in the LSPR absorbance intensity or wavelength represents the difference in particle size, shape, aggregation state, as well as free electron cloud on the surface of the nanostructure [49,52]. We used the UV-Vis spectrum to observe the optical properties of the SiO_2_@Au nanostructures modified by three kinds of ligand (11-mercaptopundecanoic acid (R-COOH), 11-mercapto-1-undecanol (R-OH), and 1-undecanethiol (R-CH_3_)). Similarly, SiO_2_@Au which removed the SiO_2_ core by HF treatment was also used as a control sample to compare its optical properties to SiO_2_@Au nanostructures. The results are shown in Appendix A and Figure 3. When R-COOH, R-CH_3_ and R-OH ligands were modified on the SiO_2_@Au surface, the absorbance intensities of all prepared SiO_2_@Au nanostructures obviously decreased (Appendix A). In addition, the extinction peak positions of SiO_2_@Au nanostructures modified by ligands were all blue-shifts (Figure 3a(i)). Maximum peak of SiO_2_@Au nanostructures modified by R-COOH ligand dramatically decreased 50 nm from 615 nm to 557 nm. Similarly, the SiO_2_@Au nanostructures modified by R-OH ligand also showed a 75 nm blue-shift to 540 nm and those modified by R-CH_3_ showed a 42 nm blue-shift to 573 nm, respectively (Figure 3b(i)). Meanwhile, the HF-treated SiO_2_@Au modified by R-COOH, R-OH, and R-CH_3_ ligands exhibited different behaviors (Figure 3a(ii)). Even though the extinction peak positions of the HF-treated SiO_2_@Au modified by R-COOH and R-OH also showed decreases in LSPR wavelength like those of SiO_2_@Au, the blue-shift differences in wavelength were small, just 15 and 18 nm for R-COOH and R-OH ligands, respectively (Figure 3b(ii)). Blue-shift decreases in wavelength of SiO_2_@Au nanostructures modified by R-COOH and R-OH are due to hydrophilic properties of OH and COOH groups of the modified nanostructure regardless of HF treatment. Interestingly, the HF-treated SiO_2_@Au modified by R-CH_3_ showed an opposite behavior compared to that of SiO_2_@Au nanostructure. While the SiO_2_@Au nanostructure modified by R-CH_3_ showed a blue-shift from 615 nm to 573 nm, the HF-treated SiO_2_@Au nanostructure modified by R-CH_3_ showed a red-shift from 546 to 561 nm. This indicates the presence of aggregation of the HF-treated SiO_2_@Au nanostructure modified by R-CH_3_. In contrast, presence of R-CH_3_ groups on the surface of SiO_2_@Au nanostructures showed little effect on the optical property of SiO_2_@Au nanostructure because of the big size of SiO_2_@Au (~211 nm) compared to HF-treated SiO_2_@Au (~8 nm). This is because of the intrinsic hydrophobic property of CH_3_ group strongly present on the surface of small Au, as shown in Figure 2c (~8 nm). The result was confirmed by replacing R-CH_3_ group by 4-aminophenol (4-ATP) with a benzene ring. Appendix A show that the extinction bands of both SiO_2_@Au with and without HF treatment were red-shifted because of the intrinsic hydrophobic aromatic ring of 4-ATP. From these results, we once again concluded that the SiO_2_@Au nanostructure possessed a stable surface for ligand modification compared to that of Au NPs without the SiO_2_ core. 

### 2.4. SERS and Peroxidase-Like Activity of SiO_2_@Au Nanostructure

The application of the SiO_2_@Au nanostructure was briefly performed in our study. In SERS measurement, we used 4-ATP as a Raman reporter to observe the SERS properties of SiO_2_@Au with and without HF treatment using a 780 nm diode pump solid-state laser. Figure 4a shows the SERS spectra of SiO_2_@Au and HF-treated SiO_2_@Au before and after incubating in 1 mM 4-ATP solution. Although several Raman signals were shown in 4-ATP untreated condition, after incubating in 1 mM 4-ATP solution, typical bands of 4-ATP were clearly obtained for both SiO_2_@Au and HF-treated SiO_2_@Au due to the electromagnetic enhancement of the decorated Au NPs. Dominant and distinct bands were seen at 1081 cm^−1^ which were assigned to C-H in-plane bending vibration. The band at 1592 cm^−1^ and 391 cm^−1^ were attributed to C-C ring and C-S stretching, respectively [32]. Interestingly, the SERS bands at 391, 1081, and 1592 cm^−1^ of SiO_2_@Au were stronger than those of HF-treated SiO_2_@Au due to the hot spot generated by Au NPs on the surface of SiO_2_@Au. These results demonstrate the importance of pre-seeding Au NPs to ensure the growth of Au NPs onto the SiO_2_ surface to generate various hot spots, enhance the electromagnetic field on or near the surface of SiO_2_@Au, and amplify the SERS signal of SiO_2_@Au compared to HF-treated SiO_2_@Au without the SiO_2_ core template.

In addition, Au NPs-based nanostructures showed peroxidase-like catalytic activities in previous reports [54,55,56]. In this study, we used the SiO_2_@Au nanostructure to catalyze for 5,5′-Tetramethylbenzidine (TMB) and hydrogen peroxide (H_2_O_2_) reaction. Figure 4b shows the UV-Vis spectra of the SiO_2_@Au with and without HF treatment. In the presence of H_2_O_2_ and under the catalysis of nanostructures, TMB was oxidized to TMB^+^, followed by conversion to TMB^2+^ in sulfuric acid (H_2_SO4) condition as mentioned in Appendix A. The presence of TMB^2+^ in reaction will show a clear absorbance peak at ~450 nm. Indeed, TMB was successfully converted to TMB^2+^ under the catalysis of the SiO_2_@Au by the presence of a clear and strong absorbance band at 453 nm (Figure 4b(i)). Similarly, the HF-treated SiO_2_@Au nanostructures could also convert TMB to TMB^2+^, as shown in in Figure 4b(ii). However, the absorbance intensity at 453 nm of the SiO_2_@Au was two times higher than that of the SiO_2_@Au with HF treatment. This is perhaps because of the easy aggregation of the HF-treated SiO_2_@Au nanostructure or because the surface of Au NPs was varied in the HF treatment as mentioned in a previous report [54].

## 3. Materials and Methods

### 3.1. Chemicals and Reagents 

Tetraethylorthosilicate (TEOS), 3-aminopropyltriethoxysilane (APTS), tetrakis(hydroxymethyl)phosphonium chloride (THPC), polyvinylpyrrolidone (PVP), phosphate buffer saline (PBS), phosphate buffer saline containing 0.1% Tween 20 (PBST), ascorbic acid (AA), chloroauric acid (HAuCl_4_), hydrofluoric acid (HF), 4-aminothiophenol (4-ATP), 4-mercaptobenzoic acid (4-MBA), 1-undecanethiol (R-CH_3_), 11-mercaptopundecanoic acid (R-COOH), 11-mercapto-1-undecanol (R-OH), hydrogen peroxide (H_2_O_2_), and 5,5′-Tetramethylbenzidine (TMB) were purchased from Sigma-Aldrich (St. Louis, MO, USA). Ethanol (EtOH), sulfuric acid (H_2_SO_4_), buffer pH 4.0, and aqueous ammonium hydroxide (NH_4_OH, 27%) were purchased from Daejung (Sihung, Gyeonggi-do, South Korea); and ultrapure water (18.2 MΩ cm) was produced by a Millipore water purification system (EXL water purification; Vivagen Co., Seongnam, Gyeonggi-do, Korea).

### 3.2. Preparation of SiO_2_@Au NPs

The SiO_2_@Au seed was prepared as in the previous report [32,33]. The SiO_2_@Au NPs were prepared by incubating Au NPs suspension (10 mL) with aminated silica NPs (2 mg) overnight. The colloids were carefully centrifuged and washed thoroughly using EtOH. The NPs were then redispersed in 2.0 mL of 1 mg/mL PVP of obtained 1 mg/mL SiO_2_@Au seed.

The SiO_2_@Au NPs were carefully prepared via the reduction and deposition of Au using AA onto SiO_2_@Au seed in PVP. Moreover, 200 µL of SiO_2_@Au (1.0 mg/mL) was briefly dispersed in 9.8 mL of water that contained 10 mg of PVP under stirring. Thereafter, 20 µL of HAuCl_4_ (10 mM) was added to the suspension, followed by the addition of 40 µL of AA (10 mM). The suspension was incubated for 15 min to completely reduce the Au^3+^ ions to Au^0^. By repeating the reduction steps, the final Au^3+^ concentration was controlled to be 150 µM. The SiO_2_@Au NPs were obtained by the centrifugation of the suspension at 8500 rpm for 15 min, then the NPs were washed thoroughly using EtOH to remove any excess reagent. The SiO_2_@Au NPs were then redispersed in 1 mL of absolute EtOH to obtain a 200 µg/mL SiO_2_@Au NP suspension.

### 3.3. Etching of Silica Core of SiO_2_@Au NPs

In order to etch the silica core of SiO_2_@Au, 500 µL of SiO_2_@Au (100 µg/mL) in PBS solution which contained 0.1% Tween 20 (PBST), was added in 500 µL of 48% HF, followed by incubation for 24 h and centrifugation for 15 min at 17,000 rpm to obtain the Au NPs suspension. The prepared Au NPs were washed thoroughly using PBST to remove any excess reagent. The Au NPs were then redispersed in 1 mL of PBST, to obtain a 50 µg/mL Au NPs suspension.

### 3.4. Separation and Dispersion of SiO_2_@Au

First, 50 µg/mL SiO_2_@Au suspension was prepared in PBST and centrifuged for 10 min at various centrifugation speeds in the range of 1000 to 18,000× *g*. The supernatant was collected and the UV-Vis spectroscopy measured in the range of 300–800 nm. The pellet was redispersed by sonication. The centrifugation and redispersion were repeated until finishing. Similarly, the separation and redispersion of 50 µg/mL Au suspension was carried out as SiO_2_@Au suspension.

### 3.5. Surface Modification of SiO_2_@Au

SiO_2_@Au was incubated with 1 mM ligands, such as 1-undecanethiol (R-CH_3_), 11-mercaptopundecanoic acid (R-COOH), and 11-mercapto-1-undecanol (R-OH) for 6 h at room temperature to modify the surface of Au NPs by CH_3_, COOH, and OH groups, respectively. Briefly, 50 µg SiO_2_@Au suspension was prepared in 500 µL EtOH. Then, 2 mM ligand solutions (500 µL) were added into SiO_2_@Au suspension and incubated for 6 h under stirring. The nanostructure was collected at 17,000 rpm for 15 min and the obtained pellet was redispersed in 1000 µL PBST to obtain an SiO_2_@Au-ligand suspension. Similarly, the surface modification of Au NPs was carried out as SiO_2_@Au NPs.

### 3.6. Peroxidase-like Catalytic Activity of SiO_2_@Au

In order to obtain the peroxidase-like catalysis of nanostructures, TMB was dissolved in EtOH to obtain a stock solution of 10 mM TMB. Similarly, a stock solution of 2 M H_2_O_2_ was freshly prepared. Thereafter, 100 µL of 10 mM TMB, 700 µL of buffer pH 4.0, 100 µL of 2 M H_2_O_2_, and 100 µL of nanostructure (200 µg/mL) were incubated for 30 min at 25 °C in vortex mixer. The mixture was added to 500 µL of 1 M H_2_SO4 to stop the reaction and incubated for 10 min for color development. The suspension was measured by UV-Vis spectroscopy at 453 nm.

### 3.7. Instrument

Tranmission electron microscope images of the sample were captured by using Libra 120 field-emission transmission electron microscope (Carl Zeiss, Oberkochen, Baden-Württemberg, Germany) with a maximum accelerated voltage of 120 kV and JEM-F200 multi-purpose electron microscope (JEOL, Akishima, Tokyo, Japan) with a maximum accelerated voltage of 200 kV. Optical properties of the sample were observed by an Optizen POP UV/Vis spectrometer (Mecasys, Seoul, Korea). Centrifugation of the sample was performed by using a Microcentrifuge 1730R (LaboGene, Lyngen, Denmark). The Raman signal of nanostructure suspension in the capillary tube was recorded by a micro-Raman system (LabRam 300, JY-Horiba, Tokyo, Japan) equipped with an optical microscope (BX41, Olympus, Tokyo, Japan). The SERS signals were collected with a back-scattering geometry using a ×10 objective lens (0.90 NA, Olympus) and with a spectrometer using a thermoelectrically cooled CCD detector. A 780 nm diode-pumped solid-state laser (CL780-100-S, CrystaLaser, Reno, NV, USA) was used as a photoexcitation source with 50 mW of laser power shot at the sample.

## 4. Conclusions

Dense Au NPs on the surface of SiO_2_@Au was successfully developed using the combination of Au seed-mediated growth in this study. The growth of Au NPs on the surface of SiO_2_ was confirmed by the color changes from light brown to dark blue, TEM images, and broad bands in the UV-vis spectra in the range of 450 nm to 750 nm with the maximum peak at 615 nm, which indicated bigger size of Au NPs on SiO_2_ surface when the Au^3+^ concentration increased to 150 µM. In addition, the prepared SiO_2_@Au nanostructure showed several advantages compared to the SiO_2_@Au treated by HF solution, such as easy separation from the solution at 3000× *g* for 10 min and stability for surface modification by R-COOH, R-OH, and R-CH_3_. Furthermore, the SiO_2_@Au nanostructure was used as a SERS substrate for 4-ATP detection and a peroxidase-like catalyst material for TMB and H_2_O_2_ reactions. As a result, the SiO_2_@Au showed a stronger SERS signal of SiO_2_@Au compared to HF-treated SiO_2_@Au. Similarly, the catalytic activity of SiO_2_@Au was two times better than that of the SiO_2_@Au with HF treatment. This study provides a potential method for nanoparticle preparation that can be replaced for Au NPs, which supports further research and development.

## Figures and Tables

**Figure 1 ijms-22-02543-f001:**
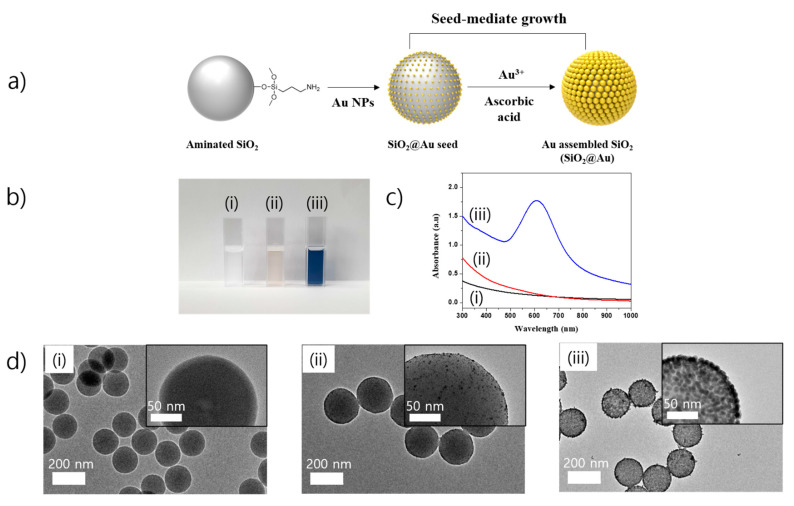
(**a**) Illustration of preparation of gold embedded silica nanostructure by using the combination of seed-growth method and reduction of chloroauric acid by ascorbic acid and polyvinylpyrrolidone. (**b**) Optical images, (**c**) UV-Vis spectra, and (**d**) Transmittance electronic microscopy (TEM) images of SiO_2_-based nanostructures (50 µg/mL): (i) SiO_2_ metal nanoparticles (NPs); (ii) SiO_2_ nanostructure (SiO_2_@Au) seed and (iii) SiO_2_@Au nanostructure.

**Figure 2 ijms-22-02543-f002:**
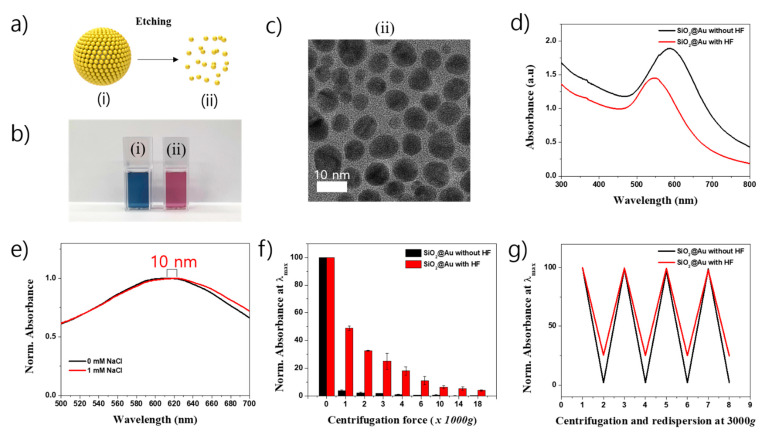
(**a**) Etching, (**b**) optical images, (**c**) TEM images, and (**d**) UV-Vis spectra of SiO_2_@Au nanostructures (50 µg/mL): (i) SiO_2_@Au without hydrofluoric acid (HF) treatment and (ii) SiO_2_@Au treated with HF. (**e**) Red-shifting of UV-Vis spectra, (**f**) centrifugation speed, and (**g**) centrifugation (2,4,6,8 in x-axis) and redispersion (3,5,7 in x-axis) of SiO_2_@Au synthesized at 150 M Au^3+^ with and without HF treatment.

**Figure 3 ijms-22-02543-f003:**
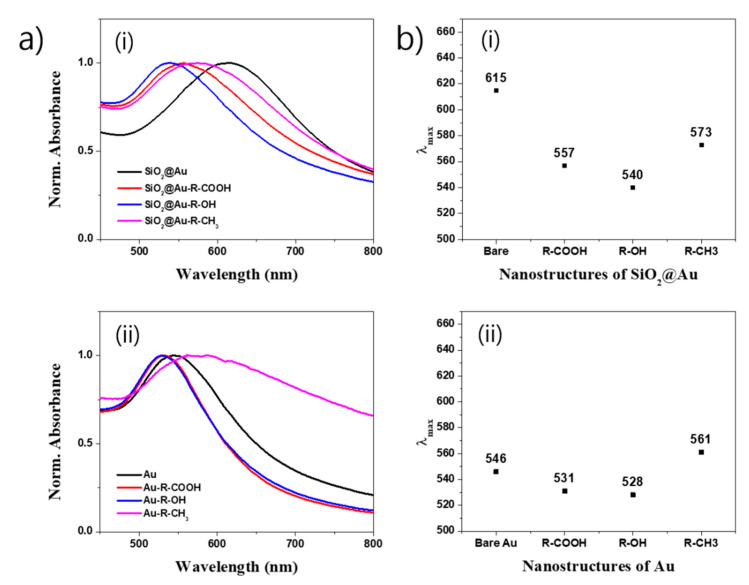
Effect of surface modification of 1-undecanethiol (R-CH_3_), 11-mercaptopundecanoic acid (R-COOH), and 11-mercapto-1-undecanol (R-OH) ligands on (**a**) the UV-Vis spectra and (**b**) plot of extinction maximum band position of SiO_2_@Au nanostructures synthesized at 150 M Au^3+^.

**Figure 4 ijms-22-02543-f004:**
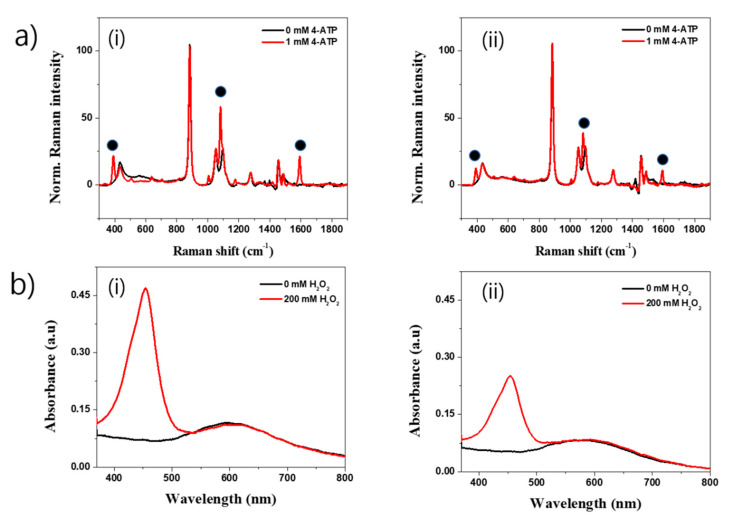
(**a**) Surface enhanced Raman scattering of SiO_2_@Au nanostructures suspended in ethanol solution and (**b**) peroxidase-like catalytic activity of SiO_2_@Au nanostructures synthesized at 150 M Au^3+^ (i) without and (ii) with HF treatment.

## Data Availability

Data is contained within the article or Appendix A.

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
