# Peer review of "Synthesis of Densely Immobilized Gold-Assembled Silica Nanostructures"

_ijms, 2021, doi:10.3390/ijms22052543_

Round 1

Reviewer 1 Report

Authors studied the synthesis of Au nanoparticles immobilized onto silica sphere and report their stability, SERS and catalytic activity. The synthesis of plasmonic nanostructures is of great interest and it is worth studying. The manuscript is well organized and data are quite clearly presented and discussed.

I have some comments to rise to improve the manuscript soundness:

  • authors synthesized SiO2@Au nanostructures at different Au+3concentration and decided to investigate the system prepared at 150uM Au+3 . a sentence to justify this choice should be added.
  • It is difficult to understand the experiment redispersion form fig 2.g. In fig 2.g is reported that after 1,3,5 and 7 centrifugation cycles the absorbance of the colloidal solutions is the same, but by the text I understand that the absorbance is the same after the redispersion. Please, clarify this point.
  • By SERS analysis, SiO2@Au nanostructures present many Raman signals even at 0-mM of 4-ATP. It seems to me that the plasmonic structure prepared have a contaminated surface and this could be a drawback of the proposed preparation. Please discuss this point.

Author Response

Dear Reviewers:

Thank you for considering the enclosed manuscript “Synthesis of densely immobilized gold-assembled silica nanostructures” (IJMS-1125303) for publication in the International Journal of Molecular Sciences as a research article.

We appreciate the comments from the reviewers who spent invaluable time and effort. We have incorporated additional modifications based on the reviewers’ thoughtful comments, which have helped us to improve the manuscript. The detailed responses to the reviewers’ comments are provided at the end of this letter.

Reviewer 2 Report

In Figure 1 and 2, the TEM magnifications are not adequate. Using a High Resolution TEM and capture the nanoparticles in 2-5 nm scale bars. The gold nanoparticles must be shown clearly.

The histogram of nano particle size must be added for all HRTEM.

The information’s of instruments must be reported in the third section.

Author Response

(The authors gave the same response as above.)

Round 2

Reviewer 2 Report

Accept new version of manuscript.